# Characterizing fermentable carbohydrate foods in the diets of children with abdominal pain-related disorders of gut-brain interaction and healthy children

Vishnu Narayana[1]☯, Jocelyn Chang[1,2]☯, Ann R. McMeans[1,2], Tasha Murphy[3], Rona L. Levy[3], Robert J. Shulman[1,2‡], Bruno P. Chumpitazi🆔[4,5‡]*

**1** Department of Pediatrics, Baylor College of Medicine, Houston, Texas, United States of America, **2** Children's Nutrition Research Center, Houston, Texas, United States of America, **3** School of Social Work, University of Washington, Seattle, Washington, United States of America, **4** Department of Pediatrics, Duke University School of Medicine, Durham, North Carolina, United States of America, **5** Duke Clinical Research Institute, Durham, North Carolina, United States of America

☯ These authors contributed equally to the work.
‡ RJS and BPC also contributed equally to the work.
* Bruno.chumpitazi@duke.edu

## Abstract

### Objectives

Restricting dietary fermentable oligosaccharides, disaccharides, monosaccharides, and polyols (**FODMAPs**) can alleviate symptoms in children with disorders of gut-brain interaction (**DGBI**). Due to the restrictions of a low FODMAP diet (**LFD**), a less restrictive FODMAP Gentle diet (**FGD**) has been suggested. However, the types and amounts of high FODMAP foods and carbohydrates commonly consumed by children have not been studied. We aimed to identify the high FODMAP foods and proportions of FODMAP carbohydrates consumed by children with DGBI and healthy children (**HC**) and to determine which usually ingested FODMAPs would be restricted on the FGD.

### Methods

Three-day diet records from both children with DGBI and HC were analyzed and compared to assess the type and amount of high FODMAP foods and carbohydrates ingested. Additionally, the ingested FODMAPs that would be restricted on the FGD were determined.

### Results

Diet records from 77 children with DGBI and 64 HC were analyzed. The number of foods ingested daily was similar between children with DGBI and HC (12.3±4.2 vs 12.9±3.4, respectively); high FODMAP foods comprised most foods eaten in

**Data availability statement:** That data may be found in the submission.

**Funding:** RJS: NIH NR013497 RJS/RLL: NIH NR016786.

**Competing interests:** I have read the journal's policy and the authors of this manuscript have the following competing interests: BPC - research funding, CAIRN diagnostics; consultant, Ironwood Pharmaceuticals; research funding, NIH RJS - research funding, CAIRN diagnostics; research funding, NIH RLL - research funding, NIH The funders had no role in study design, data collection and analysis, decision to publish, or preparation of the manuscript.

both groups. Children with DGBI (vs. HC) ate fewer high FODMAP foods per day (6.5±2.3 vs 8.7±2.4, P<0.0001, respectively). Fructans were the most consumed FODMAP carbohydrate in both groups, and children with DGBI (vs. HC) consumed fewer fructans, lactose, fructose, and polyols (all P<0.0001). The top 3 food categories consumed in both groups were wheat-containing foods, dairy, and fruits/ 100% fruit juices. In children with DGBI, 80.9% of the high FODMAP foods consumed would be limited on the FGD.

## Conclusions

Children with DGBI consume fewer high FODMAP foods and carbohydrates than HC, with the top consumed FODMAP carbohydrates being fructans, lactose, and fructose. A FGD would restrict most high FODMAP foods consumed by children with DGBI.

---

## Introduction

Abdominal pain-related disorders of gut-brain interaction (**DGBI**), such as irritable bowel syndrome, are very prevalent [1]. Contributors to symptoms include visceral hypersensitivity, psychosocial distress, gut dysmotility, abnormal gut immune function, abnormal gut microbiota composition, and diet [2]. Dietary culprits include fermentable oligosaccharides, disaccharides, monosaccharides, and polyols (**FODMAPs**), which are carbohydrates that are poorly absorbed in the small intestine [3]. After reaching the colon, FODMAPs are rapidly fermented, producing gas and other metabolites [3–5]. The gas and metabolite production, in addition to osmotically-driven water secretion into the lumen of the colon, can contribute to DGBI symptom generation [4]. Restricting FODMAP dietary intake through a low FODMAP diet (**LFD**) may ameliorate pediatric and adult DGBI symptoms, including pain, bloating, flatus, diarrhea, nausea, and heartburn [6–8].

There are challenges with implementing a LFD, including the need for appropriate educational resources, patient adherence, concern for potential adverse effects such as decreases in bacteria associated with health (e.g., *Bifidobacteria*), the potential for unintentional weight loss, and risk of developing disordered eating [9–11]. Due to these possible negative impacts, some experts do not currently recommend a LFD for children [12–15]. This has led to proposals for diets that restrict fewer fermentable carbohydrates (e.g., the "FODMAP Gentle" diet, **FGD**) in an attempt to be more pediatric-friendly [15,16]. However, determining potentially effective and less restrictive diets such as the FGD for children is hampered by lack of knowledge related to which high FODMAP foods normally are consumed by children and whether the FODMAP intake of healthy children (**HC**) differs from that of children with DGBI. This information is needed to determine the degree to which a less restrictive diet (e.g., FGD) may impact FODMAP intake. Therefore, we sought to identify the high FODMAP foods and proportions of FODMAP carbohydrates consumed by children with DGBI and HC and to determine which usually ingested FODMAPs would be restricted on the FGD.

## Materials and methods

### Study design

The data analyzed in this study were dietary intake from a large trial of children with DGBI (NCT03823742) recruited between February 22, 2019, and May 1, 2024, and the dietary intake of HC from a separate study, which recruited subjects between May 1, 2013, and October 29, 2018. The Baylor College of Medicine Institutional Review Board for Human Subjects approved both studies (protocols H-31926 and H-43391). Demographic categorization of race/ethnicity and sex was determined by the parent. Written informed consent was obtained from the study participants' parents (or guardians), and assent was obtained from the study participants. Prior to any kind of intervention, parents of participants in both studies completed a 3-day baseline food record of their child's habitual diet after receiving detailed information on how to complete the diet records. Three-day diet records are superior to 24-hour recall and 5-day food records in children based on agreement with observed and reported food intake [17].

### Study population

Participants were children 7–12 years of age from the greater Houston, Texas metropolitan area. Additional inclusion criteria for HC were: 1) Lack of any chronic health issues and 2) No previous healthcare visits related to abdominal pain. The additional inclusion criteria for children with DGBI were meeting the pediatric Rome IV definitions for an abdominal pain-related DGBI [18]. Exclusion criteria for both children with DGBI and HC included: children following specific diet alterations (e.g., LFD); those with eating disorders; any recent weight loss (≥ 5% of body weight); or organic gastrointestinal or systemic diseases (e.g., celiac disease, Crohn's disease, cancer).

Children with DGBI were recruited based on having had a visit with their pediatrician or pediatric gastroenterologist for abdominal pain. The medical record of these children was reviewed by the study team to ensure a secondary etiology was not identified. The parents of potential patients were contacted, and using a modified Rome IV questionnaire, potential participants were screened to ensure they met pediatric DGBI Rome IV criteria, including having abdominal pain at least 4 times per month without evidence of an organic abdominal pain disorder. HC were recruited via flyers and advertisements. HC also were screened to ensure they met inclusion and exclusion criteria.

### Data collection

Following enrollment, 3-day food records of foods and drinks consumed by the children were completed by the parents. Details included the name of the food, the amount eaten, and the parts of the food if the food had multiple components (e.g., a sandwich had all components listed). Diet records then were analyzed by two research dietitians well-versed in DGBI and FODMAPs and in providing LFD education. Data were entered into the University of Minnesota Nutrition Data System for Research software version 2019 [19]. Both children with DGBI and HC kept 2-week pain and stooling diaries. Diaries were used to subtype (e.g., irritable bowel syndrome) children with DGBI, and to ensure HC did not have more than one episode of abdominal pain over two weeks [20,21].

### Analysis and classification of foods

High FODMAP foods were identified using guidelines derived from the Monash University Low FODMAP dietary food guide and smartphone app [22]. Foods were considered "high FODMAP" if the amount consumed exceeded the portion size recommended from the Monash University guide and app. High FODMAP foods were further characterized by the type of FODMAP carbohydrate within the food, the food categories to which the foods belong, and whether the foods contained high fructose corn syrup as a major ingredient. Statistical comparisons between the dietary intake of the two groups were made using both chi-square for categorical data and t-testing for continuous data using IBM SPSS Statistics for Windows (IBM Corp. Released 2023. IBM SPSS Statistics for Windows, Version 29.0.2.0 Armonk, NY: IBM Corp). In

addition, based on published guidelines related to the FGD, we determined whether the high FODMAP foods consumed by children with DGBI would have been restricted [15,16].

## Results

At the time of analysis, 77 children with DGBI (35 with irritable bowel syndrome, 25 with functional abdominal pain, and 17 with incomplete data to determine subtype) had completed baseline diet records for the ongoing study. Data from 64 HC were available. Self-reported demographic data were similar between groups (Table 1).

Out of a total of 4734 foods consumed in both groups, 2792 (59.0%) were high FODMAP. Children with DGBI consumed fewer high FODMAP foods daily per child than HC (Table 2). High FODMAP foods comprised 1481 (52.5%) and 1311 (68.6%) of total foods consumed by children with DGBI and HC, respectively.

Based on the FODMAP carbohydrate, fructan-containing foods were the most numerous in the diet of both groups, followed by foods containing lactose and fructose (Table 2). Individual foods containing more than one type of FODMAP contributed 501 (33.8%) and 638 (48.7%) of the high FODMAP foods consumed by children with DGBI and HC, respectively. Children with DGBI (vs. HC) consumed significantly fewer fructans, lactose, fructose, and polyols but not galactans.

Based on food category, wheat-containing foods were the most prevalent, followed by dairy and fruits/100% fruit juices (Table 2); children with DGBI consumed significantly fewer of these foods compared to HC. Children with DGBI (vs. HC) consumed less FODMAP-containing sweet and savory snacks. There was no significant difference in the intake of beverages, condiments, vegetables, and sweeteners.

Based on the high FODMAP-containing foods, children with DGBI most frequently consumed milk, sodas, and fructose-sweetened drinks (e.g., fruit-flavored drinks containing high fructose corn syrup), whereas HC consumed milk, fresh fruit, and wheat-containing loaf bread (Table 2). High FODMAP foods containing high fructose corn syrup as a major ingredient constituted 248 (16.7%) of foods consumed by children with DGBI and 388 (29.6%) of foods consumed by HC.

In children with DGBI, 1198 (80.9%) of the consumed high FODMAP foods would be restricted on a FGD. Of the 283 high FODMAP foods permitted on a FGD, 139 (49.1%) contained high fructose corn syrup as a major ingredient. Of these, beverages and condiments (e.g., sodas, jam/jelly, ready-to-drink lemonade, pancake syrup, and flavored drinks) comprised a significant proportion.

**Table 1. Demographics of Children with DGBI versus Healthy Children.**

|  | DGBI (n = 77) | HC (n = 64) | P-value |
|---|---|---|---|
| **Age** (years ± SD) | 10.0 ± 1.7 | 9.7 ± 1.5 | 0.25 |
| **Sex** |  |  | 0.68 |
| Female, n (%) | 46 (59.7) | 36 (56.3) |  |
| Male, n (%) | 31 (40.3) | 28 (43.8) |  |
| **Race, ethnicity**, n (%) |  |  | 0.07 |
| White, non-Hispanic | 22 (28.6) | 32 (50.0) |  |
| White, Hispanic | 16 (20.8) | 14 (21.9) |  |
| Black, non-Hispanic | 23 (29.9) | 10 (15.6) |  |
| Black, Hispanic | 4 (5.2) | 2 (3.1) |  |
| Asian, non-Hispanic | 2 (2.6) | 0 (0.0) |  |
| American Indian/ Hawaiian, non-Hispanic | 0 (0.0) | 0 (0.0) |  |
| American Indian/ Hawaiian, Hispanic | 4 (5.2) | 1 (1.6) |  |
| More than one race, non-Hispanic | 1 (1.3) | 3 (4.7) |  |
| More than one race, Hispanic | 1 (1.3) | 2 (3.1) |  |
| Black, no ethnicity identified | 1 (1.3) | 0 (0.0) |  |
| Hispanic, no race identified | 3 (3.9) | 0 (0.0) |  |

**Table 2. Daily Food Consumption and High FODMAP Intake in Children with DGBI versus Healthy Children.**

| | DGBI (n=77) | HC (n=64) | P-value |
|---|---|---|---|
| **Total Foods Consumed** | 2822[1] | 1912[1] | |
| Total Foods per subject per day (mean±SD) | 12.3±4.2 | 12.9±3.4 | 0.36 |
| High FODMAP Foods per subject per day (mean±SD) | 6.5±2.3 | 8.7±2.4 | <0.001 |
| **FODMAP Sugar**, n (%)[2] | | | |
| Fructans[3] | 850 (30) | 748 (39) | <0.0001 |
| Lactose[4] | 552 (20) | 527 (28) | <0.0001 |
| Fructose[5] | 412 (15) | 570 (30) | <0.0001 |
| Polyols[6] | 82 (3) | 100 (5) | <0.0001 |
| Galacto-oligosaccharides[7] | 82 (3) | 53 (3) | 0.79 |
| HFCS[8] | 248 (9) | 388 (20) | <0.0001 |
| **Food Category**, n (%)[2] | | | |
| Wheat | 557 (20) | 534 (28) | <0.0001 |
| Dairy | 344 (12) | 342 (18) | <0.0001 |
| Fruits and 100% Fruit Juices | 168 (6) | 194 (10) | <0.0001 |
| Beverages (e.g., sodas, juice w/ high fructose corn syrup) | 144 (5) | 93 (5) | 0.71 |
| Condiments (e.g., salad dressing, mayonnaise, ketchup, jelly, jam) | 125 (4) | 74 (4) | 0.35 |
| Sweet Snacks (e.g., candy, cookies, brownies) | 125 (4) | 143 (7) | <0.0001 |
| Vegetables | 111 (4) | 80 (4) | 0.67 |
| Savory Snacks (e.g., chips) | 84 (3) | 90 (5) | 0.002 |
| Sweeteners (e.g., fructose, honey, added polyols) | 64 (2) | 35 (2) | 0.30 |
| Legumes | 42 (1) | 16 (1) | 0.06 |
| Nuts, seeds, nut/seed butters | 8 (<1) | 13 (1) | 0.04 |
| **Top High FODMAP Foods Consumed**, n (%)[2] | | | |
| Milk | 146 (5.2) | 121 (6.3) | .09 |
| Sodas[9] | 69 (2.4) | 31 (1.6) | .05 |
| Fructose-sweetened drinks[9] | 67 (2.4) | 45 (2.4) | 0.96 |
| Breaded Animal Protein[10] | 65 (2.3) | 34 (1.8) | 0.22 |
| Fresh fruit | 62 (2.2) | 74 (3.9) | <0.001 |
| Loaf bread[11] | 61 (2.2) | 63 (3.3) | 0.02 |
| Savory condiments[12] | 58 (2.1) | 18 (0.9) | <0.005 |
| 100% Fruit juice | 44 (1.6) | 46 (2.4) | 0.04 |
| Sweet/enriched bread[11] | 43 (1.5) | 51 (2.8) | <0.01 |
| Pizza[10] | 42 (1.5) | 34 (1.8) | 0.44 |
| Mixed pasta[10] | 35 (1.2) | 35 (1.8) | 0.10 |
| Cookies[11] | 34 (1.2) | 35 (1.8) | 0.08 |
| Seasonings[13] | 23 (0.8) | 41 (2.1) | <0.001 |
| Crackers[11] | 20 (0.7) | 40 (2.1) | <0.0001 |
| Fruit snacks | 20 (0.7) | 35 (1.8) | <0.0005 |

[1]Total count of all foods consumed.

[2]Percent out of total foods consumed.

[3]Fructans are found in wheat, garlic, onion, legumes, inulin, and chicory root, which are types of fiber and the main ingredient in fiber supplements or added to baked goods to increase fiber content.

[4]Lactose is found in dairy products and may be added to some food products such as chocolate bars, flavored chips, and baked sweets.

[5]Fructose is found in many fruits and some vegetables such as apples, pears, and asparagus; it is found in honey and agave syrup and may be added as fructose or high fructose corn syrup to provide a sweet flavor in different food products, such as sodas, jams, ketchup, and baked sweets.

*(Continued)*

**Table 2.** (Continued)

[6]Polyols can be found in different fruits such as peaches, plums, and avocado and may be added in different forms (i.e., sorbitol, mannitol, xylitol) to impart a sweet flavor to different food products, most notably sugar free candies and sugar free gums.

[7]Galacto-oligosaccharides can be found in beans and nuts.

[8]High fructose corn syrup; a syrup of fructose and glucose, usually found in a 55%/45% ratio in food products.

[9]Contains high fructose corn syrup.

[10]Contains wheat, onion, and/or garlic seasoning.

[11]Contains wheat and may contain high fructose corn syrup.

[12]Contains onion, garlic, and/or high fructose corn syrup.

[13]Contains dried or powdered onion and/or garlic added during at home food preparation process. Does not include packaged foods that already contain spices.

## Discussion

Determining the type of high FODMAP foods and FODMAP carbohydrates consumed by children with DGBI and HC children is important for tailoring educational content and aiding in the creation of diets that restrict FODMAP intake. We determined that high FODMAP foods comprise a majority of the foods eaten by both children with DGBI and HC. Children with DGBI (vs. HC) consumed fewer high FODMAP foods and several FODMAP carbohydrates. We also identified that a large majority of FODMAP foods normally eaten by children with DGBI would be restricted on a FGD.

We hypothesize the less frequent ingestion of high FODMAP foods by children with DGBI may relate to self-restriction (either intentional or due to unconscious behavior) to avoid worsening their GI symptoms. Supporting the former, a recent study of school children reporting bothersome GI symptoms identified an association between high FODMAP foods and worse GI symptoms [23]. Furthermore, children with IBS self-restrict more foods than healthy children [24]. Additional studies are needed to determine the reasons why children with DGBI eat fewer FODMAPs than HC.

To our knowledge, this is the first study to describe FODMAP intake in children with DGBI and HC under normal dietary circumstances. Fructans were the FODMAP carbohydrate ingested most commonly in both groups. A U.S. Department of Agriculture survey from 1994–1996 estimated that 75% of fructan intake in individuals was from wheat and 25% from onions [25]. Our study findings are in agreement with the survey data, as wheat-containing products, particularly breaded meat products such as chicken nuggets and wheat bread, were the primary contributors to fructan intake for both groups. Onion and garlic powders via seasoning contributed to fructan intake to a lesser degree. Some of the consumed foods containing onion and garlic powder seasonings included flavored chips, processed and seasoned meats, and seasoned beans (i.e., refried beans). Fructans, which arrive essentially intact into the colon after consumption, are often a focus of a low FODMAP diet as fructans have been found to induce more DGBI symptoms than other FODMAPs [26]. Given our finding that wheat, which contains fructans, is the most commonly ingested category of food, particular attention should be paid to the restriction of wheat products on a reduced FODMAP diet.

Lactose and fructose were the second and third most commonly ingested FODMAPs. Lactose malabsorption has previously been identified as a potential dietary factor contributing to DGBI symptoms [27]. The prevalence of lactose intolerance, as opposed to lactose malabsorption, appears to be increased in those with DGBI, which may be related to visceral hyperalgesia (increased gut sensitivity to stimuli) [28]. These data suggest that emphasis on reduction in lactose and fructose would be high-yield on a reduced FODMAP diet in children with DGBI. As some of the products containing lactose or fructose (particularly high fructose corn syrup) may be unhealthy, future studies related to FODMAP dietary education may look to include additional facets such as regulation of craving training [29].

The FGD has been proposed as an alternative, pediatric-friendly, and less restrictive version of the traditional LFD [15,16]. We found that the vast majority of high FODMAP foods commonly consumed by children with DGBI would be

restricted on a FGD. One particular advantage of the FGD may be its relative ease related to education and adherence, as only 15 different food and/or food types (e.g., wheat) are restricted. Our data suggest that a FGD would be successful in restricting a large number of high FODMAP foods consumed by children. However, FGD efficacy in children with DGBI remains to be determined.

Of the high FODMAP foods consumed by the children in our study that would be allowed on a FGD, about half contained high fructose corn syrup as a major ingredient - not surprising given its prevalence in many foods in the US. Important to note is that the FGD was developed in Australia, where high fructose corn syrup is not added to food as frequently as it is in the US [16]. Thus, in the US, restriction of foods containing this FODMAP (e.g., sodas, jams) is likely needed when children are placed on a FGD.

There are some limitations to our study. Parents may not have accurately recorded the food intake of their children with the use of food records. However, 3-day food records are considered a valid method to record food consumption and are considered a better method of food capture compared to other approaches, such as 24-hour recall [17]. Children were recruited from a single geographic area, and diets may differ based on geographic location. However, our study population included a racially and ethnically diverse population, which may increase the generalizability of our findings. Further studies in different geographic areas can be conducted to confirm our findings. Finally, there is a time difference between the diary capture of children with DGBI and HC, and it is possible that information related to FODMAPs or other nutritional recommendations may have influenced the dietary habits of children with DGBI. However, we note that the majority of foods eaten by both groups were high FODMAP foods.

A major strength of our study is that, to our knowledge, it is the first to characterize and compare the FODMAP intake of children with DGBI versus HC. Also, the two populations were well-matched in terms of demographics, increasing the rigor of the findings. Additionally, the dietitians participating in the study had experience with analyzing food records for high FODMAP foods and maintained high quality in their review, including follow-up with families, if needed, regarding specific food record entries.

## Conclusions

In summary, both children with DGBI and HC consume a significant amount of high FODMAP foods, though children with DGBI consume relatively less. Fructans, lactose, and fructose are the most consumed FODMAP carbohydrates via wheat-containing foods, dairy, and fruits/100% fruit juices. The FGD restricts a large majority of the high FODMAP foods consumed by children with DGBI.

## Author contributions

**Conceptualization:** Vishnu Narayana, Jocelyn Chang, Ann R. McMeans, Rona L. Levy, Robert J. Shulman, Bruno P. Chumpitazi.

**Data curation:** Vishnu Narayana, Jocelyn Chang, Ann R. McMeans, Tasha Murphy, Bruno P. Chumpitazi.

**Formal analysis:** Vishnu Narayana, Jocelyn Chang, Ann R. McMeans, Bruno P. Chumpitazi.

**Funding acquisition:** Rona L. Levy, Robert J. Shulman.

**Investigation:** Vishnu Narayana, Jocelyn Chang.

**Methodology:** Vishnu Narayana, Jocelyn Chang, Ann R. McMeans, Robert J. Shulman, Bruno P. Chumpitazi.

**Project administration:** Jocelyn Chang, Tasha Murphy, Rona L. Levy, Robert J. Shulman, Bruno P. Chumpitazi.

**Resources:** Rona L. Levy, Robert J. Shulman.

**Supervision:** Rona L. Levy, Bruno P. Chumpitazi.

**Validation:** Ann R. McMeans.

**Writing – original draft:** Jocelyn Chang, Ann R. McMeans.

**Writing – review & editing:** Vishnu Narayana, Jocelyn Chang, Tasha Murphy, Rona L. Levy, Robert J. Shulman, Bruno P. Chumpitazi.

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
