## [Decision Letter · Decision Letter 0]

30 Oct 2024

PONE-D-24-38938Characterizing Highly Fermentable Carbohydrate Foods in the Diets of Children with Disorders of Gut-Brain Interaction and Healthy ChildrenPLOS ONE

Dear Dr. Chumpitazi,

Thank you for submitting your manuscript to PLOS ONE. After careful consideration, we feel that it has merit but does not fully meet PLOS ONE’s publication criteria as it currently stands. Therefore, we invite you to submit a revised version of the manuscript that addresses the points raised during the review process.

We look forward to receiving your revised manuscript.

Kind regards,

Edwin Hlangwani

Academic Editor

PLOS ONE

“RJS: NIH NR013497

RJS/RLL: NIH NR016786”

Reviewers' comments:

Reviewer's Responses to Questions

**Comments to the Author**

1. Is the manuscript technically sound, and do the data support the conclusions?

Reviewer #1: Yes

Reviewer #2: Partly

2. Has the statistical analysis been performed appropriately and rigorously? 

Reviewer #1: Yes

Reviewer #2: Yes

3. Have the authors made all data underlying the findings in their manuscript fully available?

Reviewer #1: Yes

Reviewer #2: Yes

4. Is the manuscript presented in an intelligible fashion and written in standard English?

Reviewer #1: Yes

Reviewer #2: Yes

5. Review Comments to the Author

Reviewer #1: The study “ Characterizing Highly Fermentable Carbohydrate Foods in the Diets of Children with Disorders of Gut-Brain Interaction and Healthy Children”

Issues that have to be clarified.

1/ The study group consists of children with abdominal pain-related disorders of gut-brain interaction (DGBI).

The authors state that :

Row 104:" Both children with DGBI and HC kept 2-week pain and stooling diaries as previously reported to ensure they met entry criteria"

What were the inclusion criteria for healthy children?

What were the criteria for the study group regarding the 2-week pain and stooling diaries? The authors refer to a study covering only IBS (18, 19).

2/ Although the diagnosis of DGBI is based on meeting the Rome IV criteria, organic causes must be excluded in many patients. The Authors do not report how children with DGBI were recruited or, most importantly, how the diagnosis of DGBI was established.

3/ DGBI encompass a broad spectrum of conditions such as IBS, functional dyspepsia, and functional abdominal pain syndrome.

According to Roman Criteria IV, distinguishing between different types of FAPD for clinical and research purposes is essential.

The study does not provide any clinical characteristics of patients in the DGBI group. Were these exclusively IBS patients? What was the rate of different abdominal pain-related disorders?

4/ The study does not explain how healthy children were recruited.

5/ There is a long time between starting the recruitment of healthy children and completing the recruitment of children with DGBI. Any nutrition recommendations published during this period could have influenced the differences in the diet composition between the studied groups.

5/ The main finding is that children with DGBI consumed fewer high-FODMAP foods. Were the children and parents/caregivers familiar with the diagnosis of DGBI? It cannot be ruled out that the lower FODMAP intake was a consequence of the diagnosis of functional abdominal pain disorder.

6/ The title should be changed to:

“ Characterizing Highly Fermentable Carbohydrate Foods in the Diets of Children with Abdominal Pain Related Disorders of Gut-Brain Interaction and Healthy Children”

Reviewer #2: I am writing to submit my review of the manuscript entitled “Characterizing Highly Fermentable Carbohydrate Foods in the Diets of Children with Disorders of Gut-Brain Interaction and Healthy Children" for your consideration. Overall, I find the manuscript's findings intriguing and the information provided useful for researchers and academia. The article has the potential to make a significant contribution to the related discipline.

However, I have some concerns regarding the clarity, detail, and precision of different sections, which I outline below:

I recommend that the authors address these concerns and provide a revised version of the manuscript for further consideration

• Please check the abstract formatting as per journal instruction

• What the words “highly” indicates? can we say simply fermentable carbohydrates ?“How your study is different from this study? "Dietary betaine supplementation improves growth performance, digestive function, intestinal integrity, immunity, and antioxidant capacity of yellow-feathered broilers." Italian Journal of Animal Science 20.1 (2021): 1575-1586.

• L-31 Is there any scientific logic for three day selction?????????Three-day diet records from both HC and children with DGBI were analyzed to assess 31 the type of high FODMAP foods and carbohydrates ingested.

• L-54 Recheck this sentences- Dietary culprits include highly 53 fermentable oligosaccharides, disaccharides, monosaccharides, and polyols 54 (FODMAPs) which are carbohydrates that are poorly absorbed in the small intestine

• All the factors can be combined in onse paragraph/line- Some patients have reported cost of the diet to be a 64 barrier to adherence.(10) In addition, given the restrictive nature of a LFD, there is 65 concern for potential adverse effects such as decreases in bacteria associated with health 66 (e.g., Bifidobacteria), the potential for unintentional weight loss, and risk of developing 67 disordered eating. Due to these possible negative impacts, some experts do not currently 68 recommend a LFD for children.(11-14)

• Cite the following in introduction section- doi: https://doi.org/10.1016/j.jep.2023.116503, doi: https://doi.org/10.1016/j.ijbiomac.2024.135063, doi: https://doi.org/10.1016/j.neuroimage.2024.120740

• How the pre mapping for subject selection was carried out?Results of initialmscreening??????? Any specific reson for selecting this area??????????/e9 Study Population 90 Participants were children 7-12 years of age from the greater Houston, Texas 91 metropolitan area

• L-171 High FODMAP foods containing high fructose corn syrup 171 as a major ingredient were consumed by 16.7% (248) of children with DGBI and 29.6% 172 (388) of HC (Revise)

• Need more clarity- After fructans, lactose and fructose were the most commonly ingested FODMAPs. 209 Lactose malabsorption is not more common in those with DGBI than HC but lactose 210 intolerance appears to be; likely related to visceral hyperalgesia commonly found in those 211 with DGBI.(25) Similar to education regarding wheat ingestion, our data would suggest 212 emphasis also should be given to a reduction in fructose-sweetened drinks including 213 sodas and 100% fruit juice in children with DGBI. In contrast, a global reduction in fresh 214 fruit might not be warranted given its value as an important component of a healthy diet.

• The discussion section could be improved by providing more context and background from following references,

doi: https://doi.org/10.1111/aphw.12522

doi: https://doi.org/10.1140/epjs/s11734-024-01161-y

doi: 10.1007/s10620-021-06831-8

doi: 10.3389/fnut.2022.1024722

• Italic all the scientific names,

• Remove grammatical mistakes

• Need to rewrite the conclusion

 Recheck Legends description is as per figure number and discussion-

 I urge the authors to improve the English language for better flow of literature.

 Please check reference style throughout

6. PLOS authors have the option to publish the peer review history of their article (what does this mean? ). If published, this will include your full peer review and any attached files.

**Do you want your identity to be public for this peer review?** For information about this choice, including consent withdrawal, please see our Privacy Policy .

Reviewer #1: No

Reviewer #2: **Yes: ** Dr. Muhammad Afzaal

---

## [Author Response · Author response to Decision Letter 0]

28 Dec 2024

Reviewer #1: The study „ Characterizing Highly Fermentable Carbohydrate Foods in the Diets of Children with Disorders of Gut-Brain Interaction and Healthy Children”

Issues that have to be clarified.

1/ The study group consists of children with abdominal pain-related disorders of gut-brain interaction (DGBI).

The authors state that :

Row 104:" Both children with DGBI and HC kept 2-week pain and stooling diaries as previously reported to ensure they met entry criteria"

What were the inclusion criteria for healthy children?

We thank the reviewer for his/her comments. Healthy children did not have any chronic health issues and could not have any healthcare visits related to abdominal pain. We have more clearly delineated inclusion criteria in the manuscript (page 6).

What were the criteria for the study group regarding the 2-week pain and stooling diaries? The authors refer to a study covering only IBS (18, 19).

We have expanded the information related to the use of the diaries (page 7-8).

2/ Although the diagnosis of DGBI is based on meeting the Rome IV criteria, organic causes must be excluded in many patients. The Authors do not report how children with DGBI were recruited or, most importantly, how the diagnosis of DGBI was established.

We have now added this information to the manuscript (page 7).

3/ DGBI encompass a broad spectrum of conditions such as IBS, functional dyspepsia, and functional abdominal pain syndrome.

According to Rome Criteria IV, distinguishing between different types of FAPD for clinical and research purposes is essential.

The study does not provide any clinical characteristics of patients in the DGBI group. Were these exclusively IBS patients? What was the rate of different abdominal pain-related disorders?

We thank the reviewer for this question and have now provided the subtypes in the manuscript (page 9).

4/ The study does not explain how healthy children were recruited.

This information has been added.

5/ There is a long time between starting the recruitment of healthy children and completing the recruitment of children with DGBI. Any nutrition recommendations published during this period could have influenced the differences in the diet composition between the studied groups.

We agree this is a limitation and have added it to the manuscript (page 15).

5/ The main finding is that children with DGBI consumed fewer high-FODMAP foods. Were the children and parents/caregivers familiar with the diagnosis of DGBI? It cannot be ruled out that the lower FODMAP intake was a consequence of the diagnosis of functional abdominal pain disorder.

While we do not know if the families of children with DGBI were familiar with the diagnosis, we previously hypothesized in the Discussion that those children with abdominal pain may have self-limited FODMAP intake in recognition of their symptoms potentially being exacerbated (page 13). Our clinical experience is that it is rare for children with DGBI seen in the clinic to have tried FODMAP restriction.

6/ The title should be changed to:

„ Characterizing Highly Fermentable Carbohydrate Foods in the Diets of Children with Abdominal Pain Related Disorders of Gut-Brain Interaction and Healthy Children”

We agree and have made the change with the additional removal of the word highly as recommended by Reviewer #2 (see below).

Reviewer #2: I am writing to submit my review of the manuscript entitled “Characterizing Highly Fermentable Carbohydrate Foods in the Diets of Children with Disorders of Gut-Brain Interaction and Healthy Children" for your consideration. Overall, I find the manuscript's findings intriguing and the information provided useful for researchers and academia. The article has the potential to make a significant contribution to the related discipline.

However, I have some concerns regarding the clarity, detail, and precision of different sections, which I outline below:

I recommend that the authors address these concerns and provide a revised version of the manuscript for further consideration

We thank the Reviewer for his/her critique and recommendations.

• Please check the abstract formatting as per journal instruction

We have checked and revised the abstract formatting.

• What the words “highly” indicates? can we say simply fermentable carbohydrates ?

We agree and have made the change to the title removing the word highly.

“How your study is different from this study? "Dietary betaine supplementation improves growth performance, digestive function, intestinal integrity, immunity, and antioxidant capacity of yellow-feathered broilers." Italian Journal of Animal Science 20.1 (2021): 1575-1586.

Our study is very different than the cited study. Our study evaluates children with DGBI and healthy children related to dietary fermentable carbohydrate consumption, and is not related to betaine (which is a chemical compound and not a fermentable carbohydrate) or supplementation or broilers.

• L-31 Is there any scientific logic for three day selection?????????Three-day diet records from both HC and children with DGBI were analyzed to assess 31 the type of high FODMAP foods and carbohydrates ingested.

Three-day diet records are superior to 24-hour recall and 5-day food records in children based on agreement with observed and reported food intake (PMID: 8195550). We have added this reference to the manuscript (page 15).

• L-54 Recheck this sentences- Dietary culprits include highly 53 fermentable oligosaccharides, disaccharides, monosaccharides, and polyols 54 (FODMAPs) which are carbohydrates that are poorly absorbed in the small intestine

We have removed the word highly (page 4). These carbohydrates are poorly absorbed in the small intestine.

• All the factors can be combined in one paragraph/line- Some patients have reported cost of the diet to be a 64 barrier to adherence.(10) In addition, given the restrictive nature of a LFD, there is 65 concern for potential adverse effects such as decreases in bacteria associated with health 66 (e.g., Bifidobacteria), the potential for unintentional weight loss, and risk of developing 67 disordered eating. Due to these possible negative impacts, some experts do not currently 68 recommend a LFD for children.(11-14)

We have shortened the paragraph as requested by the Reviewer (page 4).

• Cite the following in introduction section- doi: https://doi.org/10.1016/j.jep.2023.116503, doi: https://doi.org/10.1016/j.ijbiomac.2024.135063, doi: https://doi.org/10.1016/j.neuroimage.2024.120740

Respectfully, we are unable to cite these references as we are unsure how these topics are related to the focus of this manuscript. The first relates to asthma and evaluation of Tingli Dazao Xiefei Decotion in an animal model. The second relates to evaluation of Lactarius deliciosus in an animal model. The third relates to brain network dynamics in childhood based on functional magnetic resonance imaging in children aged 2 to 8 years.

• How the pre mapping for subject selection was carried out?Results of initial screening??????? Any specific reason for selecting this area??????????/e9 Study Population 90 Participants were children 7-12 years of age from the greater Houston, Texas 91 metropolitan area

Please see responses to Reviewer #1 related to participant recruitment, inclusion criteria, exclusion criteria, and screening. Houston, Texas is an ideal location for clinical studies given its recognition as one of the most racially/ethnically

diverse cities in the United States, increasing the generalizability of our findings.

• L-171 High FODMAP foods containing high fructose corn syrup 171 as a major ingredient were consumed by 16.7% (248) of children with DGBI and 29.6% 172 (388) of HC (Revise)

We have revised this sentence (page 12).

• Need more clarity- After fructans, lactose and fructose were the most commonly ingested FODMAPs. 209 Lactose malabsorption is not more common in those with DGBI than HC but lactose 210 intolerance appears to be; likely related to visceral hyperalgesia commonly found in those 211 with DGBI.(25) Similar to education regarding wheat ingestion, our data would suggest 212 emphasis also should be given to a reduction in fructose-sweetened drinks including 213 sodas and 100% fruit juice in children with DGBI. In contrast, a global reduction in fresh 214 fruit might not be warranted given its value as an important component of a healthy diet.

We thank the Reviewer, and have edited the paragraph for clarity (page 14).

• The discussion section could be improved by providing more context and background from following references,

doi: https://doi.org/10.1111/aphw.12522 (regulation of craving training)

doi: https://doi.org/10.1140/epjs/s11734-024-01161-y (brain networks in autism)

doi: 10.1007/s10620-021-06831-8 (innate lymphoid cells in inflammatory bowel disease)

doi: 10.3389/fnut.2022.1024722 (essential oil on lipid metabolism and gut microbiota)

We thank the Reviewer and have summarized the topics in parentheses next to the references above. We were able to cite the reference related to the regulation of craving training in the manuscript (page 14). However, respectfully, as the focus of the manuscript did not overlap with those in the other references, we did not cite the other references.

• Italic all the scientific names,

• Remove grammatical mistakes

• Need to rewrite the conclusion

§ Recheck Legends description is as per figure number and discussion-

§ I urge the authors to improve the English language for better flow of literature.

§ Please check reference style throughout

We have followed the above instructions with edits throughout the manuscript.

---

## [Editor Report · Decision Letter 1]

14 Jan 2025

Characterizing Fermentable Carbohydrate Foods in the Diets of Children with Abdominal Pain Related Disorders of Gut-Brain Interaction and Healthy Children

PONE-D-24-38938R1

Dear Dr. Chumpitazi,

We’re pleased to inform you that your manuscript has been judged scientifically suitable for publication and will be formally accepted for publication once it meets all outstanding technical requirements.

Kind regards,

Edwin Hlangwani

Academic Editor

PLOS ONE

Additional Editor Comments (optional):

In the author's "Response to Reviwers", the response "Please see responses to Reviewer #1 related to participant recruitment, inclusion criteria, exclusion criteria, and screening. Houston, Texas is an ideal location for clinical studies given its recognition as one of the most racially/ethnically diverse cities in the United States, increasing the generalizability of our finding" to one of the comments raised by Reviewer 2 is never ideal. Please avoid responding in that manner in the future. It may be better to copy the same response to the appropriate section.
---

## [Editor Report · Acceptance letter]

PONE-D-24-38938R1

PLOS ONE

Dear Dr. Chumpitazi,

I'm pleased to inform you that your manuscript has been deemed suitable for publication in PLOS ONE. Congratulations! Your manuscript is now being handed over to our production team.

Kind regards,

on behalf of

Dr. Edwin Hlangwani

Academic Editor

PLOS ONE